# Effects of Low and High Maternal Protein Intake on Fetal Skeletal Muscle miRNAome in Sheep

**DOI:** 10.3390/ani14111594

**Published:** 2024-05-28

**Authors:** Bilal Akyüz, Md Mahmodul Hasan Sohel, Yusuf Konca, Korhan Arslan, Kutlay Gürbulak, Murat Abay, Mahmut Kaliber, Stephen N. White, Mehmet Ulas Cinar

**Affiliations:** 1Department of Genetics, Faculty of Veterinary Medicine, Erciyes University, Kayseri 38039, Türkiye; bakyuz@erciyes.edu.tr (B.A.); sohel.mmh@gmail.com (M.M.H.S.); korhanarslan@erciyes.edu.tr (K.A.); 2Genome and Stem Cell Centre, Erciyes University, Kayseri 38039, Türkiye; 3Department of Animal Science, Faculty of Agriculture, Erciyes University, Kayseri 38039, Türkiye; yusufkonca@erciyes.edu.tr (Y.K.); mkaliber@erciyes.edu.tr (M.K.); 4Department of Obstetrics and Gynecology, Faculty of Veterinary Medicine, Erciyes University, Kayseri 38039, Türkiye; kgurbulak@erciyes.edu.tr (K.G.); mabay@erciyes.edu.tr (M.A.); 5Department of Veterinary Microbiology & Pathology, Washington State University, Pullman, WA 99164, USA; stephen_white@wsu.edu

**Keywords:** *Ovis aries*, nutrimiromics, dietary protein, prenatal life, gestation

## Abstract

**Simple Summary:**

Simple Summary: Prenatal maternal feeding is critical for fetal development. miRNAs may impart unique metabolic and immunologic effects on developing fetuses. The fetal muscle miRNAome was found to be altered via maternal protein intake levels. oar-miR-3957-5p and miR-329a-3p were up-regulated in low- and standard-protein diets vs. high-protein diets. Many of the putative miRNA target genes and genetic pathways identified were involved in known muscle disease and immune function.

**Abstract:**

Prenatal maternal feeding plays an important role in fetal development and has the potential to induce long-lasting epigenetic modifications. MicroRNAs (miRNAs) are non-coding, single-stranded RNAs that serve as one epigenetic mechanism. Though miRNAs have crucial roles in fetal programming, growth, and development, there is limited data regarding the maternal diet and miRNA expression in sheep. Therefore, we analyzed high and low maternal dietary protein for miRNA expression in fetal *longissimus dorsi*. Pregnant ewes were fed an isoenergetic high-protein (HP, 160–270 g/day), low-protein (LP, 73–112 g/day), or standard-protein diet (SP, 119–198 g/day) during pregnancy. miRNA expression profiles were evaluated using the Affymetrix GeneChip miRNA 4.0 Array. Twelve up-regulated, differentially expressed miRNAs (DE miRNAs) were identified which are targeting 65 genes. The oar-3957-5p miRNA was highly up-regulated in the LP and SP compared to the HP. Previous transcriptome analysis identified that integrin and non-receptor protein tyrosine phosphatase genes targeted by miRNAs were detected in the current experiment. A total of 28 GO terms and 10 pathway-based gene sets were significantly (*p*_adj_ < 0.05) enriched in the target genes. Most genes targeted by the identified miRNAs are involved in immune and muscle disease pathways. Our study demonstrated that dietary protein intake during pregnancy affected fetal skeletal muscle epigenetics via miRNA expression.

## 1. Introduction

In mammals, maternal nutrition in pregnancy plays a critical role in intrauterine development, fetal survival, growth, and organogenesis [1,2]. Therefore, maternal under- or over-nutrition during gestation may cause low offspring growth and performance [3] with significant influence on the economic viability of the livestock industry [4,5].

Dietary nutrients, especially macronutrients, can influence the rate of gene transcription directly through interactions with the genomic regulatory elements or indirectly affecting and/or modulating important signaling pathways [6,7]. The understanding of how nutrition affects an individual’s phenotype has greatly benefited from developments in molecular biology and genomics, and this area is called nutrigenomics. The term “epigenetics” describes heritable phenotypic alterations that occur without a change in the DNA sequence [8]. A few examples of epigenetic mechanisms are those that modify the chromatin structure, such as DNA methylation and histone modifications, as well as those that control the activity of proteins, like microRNAs (miRNAs). The term “Nutrimiromics” has been used to refer to the study of interactions between nutrients and food components with the organism’s genome via modification of gene expression due to epigenetic processes related to miRNAs [9]. Due to an incomplete understanding of how nutrition affects fetal growth regulation mechanisms, intrauterine growth restriction (IUGR) persists as a significant issue in animal husbandry despite ongoing advancements in herd management and extensive research using contemporary analytical techniques into mammalian nutritional requirements [10,11,12]. An improperly balanced diet is the main culprit when it comes to livestock. It is interesting to note that intrauterine growth inhibition syndrome can result from either a high-protein or low-protein diet [13]. According to Meza-Herrera et al. [14], one of the primary causes of this occurrence is the delay in blastocyst development, brought on by the higher levels of hazardous protein breakdown products in the mother’s bloodstream. A high-protein diet also raises the energy costs associated with hepatic gluconeogenesis and urine output, which limits the fetus’s access to energy-rich substances. High-protein intake and elevated ammonia levels (up to 300 percent higher than the controls) in pregnant rats and sheep have been linked to reduced blastocyst development, impaired embryo metabolism, and fetal growth abnormalities [15,16]. When a pregnant woman consumes a high-protein diet, her amino acid profile shifts, and her concentrations of threonine, glutamine, glycine, alanine, and serine drop, which slows the formation of fetal tissue [17]. A high-protein diet was found to have an impact on the GH/JAK/STAT/IGF pathway in mice. This led to a reduction in placental growth hormone levels, which, in turn, prevented the fetus from gaining weight [18]. Reducing the amount of protein consumed through diet has the impact of decreasing the number of amino acids present in the mother’s blood plasma, the fetus’s blood plasma, and uterine fluid. According to Wu et al. [19], this ultimately results in the suppression of embryonic growth. Protein intake has a variety of nutritional and biological effects. Protein plays a vital role in the regulation of food intake, glucose and lipid metabolism, blood pressure, bone metabolism, and immunological function in addition to nutritional importance as a supply of amino acids for protein synthesis [20]. Proteins’ physiological functions are influenced by their physico-chemical characteristics, amino acid make-up, and bioactive peptides encoded into their amino acid sequences [20]. Furthermore, research has been conducted on how proteins affect fetal and post-natal development [21]. To date, more experiments have focused on protein restriction compared to high protein regarding the maternal nutritional effects on the fetus and infant [22,23]. Blumfield and Collins [23] stated that an altered maternal protein intake during pregnancy might have detrimental effects on the offspring, and an optimal macronutrient ratio is crucial for the pre- and post-natal environments.

Additionally, protein content is one of the most expensive and limiting feed characteristics for livestock nutrition. As a feed ingredient, protein content is usually provided from human-edible ingredients like soybean, other oilseeds, and grains [24]. Thus, careful management of these limiting feedstuffs is critical for sustainable agriculture in the context of a growing world population and increasing livestock product demand [25]. Furthermore, balanced protein consumption may help to control nitrogen pollution to aid both environmental and economic sustainability [26].

Previous research has examined under-nutrition or over-nutrition conditions to test the effects of maternal nutrition on fetal development in livestock and model animals, including the fetal programming of muscle and fat tissues [22,27,28]. However, few studies have examined the potential effects of maternal protein intake on the overall miRNA expression of fetal skeletal muscle. We have previously reported that maternal diets differing in protein content alter the fetal skeletal transcriptome, and that maternal high-protein diets, compared to standard-protein diets, trigger pathways related to the immune system and disease in sheep skeletal muscle [28]. On the other hand, the mechanisms by which these diets affect fetal miRNAome in sheep skeletal muscle are not known. The objective of this study was to investigate the impact of different maternal dietary protein levels during mid-to-late gestation on the total miRNA expression of fetal skeletal muscle in sheep, specifically to determine the consequences of the maternal dietary protein intake on fetuses throughout gestation on (1) the expression of whole genome skeletal muscle miRNAs; (2) biological pathways and gene ontology related to the miRNAome; and (3) the understanding of the connections between miRNAs and their targets.

## 2. Materials and Methods

### 2.1. Experimental Animals and Diets

Details of the ewes and diets were described previously [28]. Briefly, three distinct protein diets (standard-, high-, and low-protein) were used in the study to assess the effect of maternal dietary protein on the miRNAome of fetal skeletal muscle. Twelve Akkaraman ewes aged two were divided into three treatment groups at random, with four ewes in each group. Ewe and fetus specifications are presented in Appendix A. After the empty ewes selected for the trial were brought from the pasture, an adaptation process was applied to get them accustomed to pen feeding. During the 15-day adaptation period, the ewe’s daily dry matter (DM) feed consumption was assessed and compared with the US National Research Council (NRC) recommendations for ewes [29]. Following the adaptation period, synchronization and insemination were applied to the empty ewes. The ewes were fed a standard diet from day 0 to 30 of gestation according to the NRC recommendations [29]. Three alternative feed regiments with variable protein content were created, consisting of wheat straw, alfalfa, soybean meal, barley, and maize grain; alfalfa and soybean meal were the main protein sources, whereas barley and corn were the main sources of energy. Wheat straw and alfalfa were additional sources of cellulose and energy. The protein content of the diets was changed by substituting different amounts of soybean meal and alfalfa. During mid-to-late gestation (Day 31 to 105 of gestation), the ewes were fed the following diets: The standard-protein diet (SP) was a conventional diet with a NRC level of the nutritional content of feeds (the daily DM intake ranged from 1252 to 1862 g, crude protein ranged from 119 to 198 g, and kcal ME/kg ranged from 2506 to 3909 g); a low-protein diet (LP) (a daily DM intake of 1247–1726 g, 73–112 g crude protein, and 2520–3649 kcal ME/kg); and a high-protein diet (HP) (the daily DM intake ranged from 1252–1629, 160–271 g crude protein, and 2513–3880 kcal ME/kg). The chemical composition of the materials for the feed was ascertained in the feed analysis laboratory in accordance with AOAC (1990). Diets were made in an isoenergetic fashion. Specific information about the diets was reported previously [28].

### 2.2. Estrus Synchronization, Semen Preparation, and Artificial Insemination

During the annual mating season, estrus synchronization was achieved using progesterone-infused pessaries (EaziBreed CIDR; 1.38 mg progesterone; Zoetis, İstanbul, Türkiye) for 12 days, followed by 2 mL of PGF2 (5 mg/mL Dinoprost tromethamine; Dinolytic; Zoetis, İstanbul, Türkiye) treatment on day 12, and 400 IU of eCG (Folligon, 1000 IU/vial PMSG, MSD Animal Health İstanbul, Türkiye) at the time of pessary removal. The semen was extracted from four 1- to 3-year-old rams using the artificial vaginal procedure in the presence of estrus ewe. A phase-contrast microscope with a heated plate set at 37 °C was used to measure the motility of the artificial vaginal method-obtained sperm. For the insemination of ewes, the ejaculates of four rams with a motility of 70% or higher were combined. To achieve homogeneity, the appropriate ejaculates were blended based on their spermatological motility. The density of the mixed ejaculates was determined using the hemocytometry method. To create 0.5 mL of semen containing 150 × 10^6^ motile spermatozoa, the semen was diluted with Trisma base, consisting of d-fructose, citric acid, and skimmed milk [30] and kept in a 35 °C water bath for further use. Using diluted fresh semen, artificial insemination was carried out intravaginally 48 h following sponge removal.

### 2.3. Necropsy, Sample Collection, and Isolation of miRNA

On the 105th day of pregnancy, the dams had necropsies. A total of 14 fetuses were gathered from 12 dams. To avoid growth bias resulting from multiple births, twin birth fetuses were not included in the experiment. There were two twin pregnancies in the LP and SP groups, respectively. To make sure that each experimental group contained three fetuses, a fetus from the HP group was also removed. A sample of muscle tissue was taken from the fetal left *longissimus dorsi* muscles. Prior to miRNA extraction, all tissues were snap-frozen in liquid nitrogen and stored at −80 °C. Thirty milligrams of each skeletal muscle tissue were homogenized in QIAzol lysis reagent (Qiagen, Hilden, Germany) with liquid nitrogen using a homogenizer (Precellys Bertin, Montigny-le-Bretonneux, France). Using a miRNeasy mini kit (Qiagen, Hilden, Germany) in accordance with the manufacturer’s instructions, total RNA containing miRNA was isolated from the tissue homogenates. The manufacturer’s recommendations were followed for the on-column DNA digestion stage using the RNase-Free DNase Set (Qiagen, Hilden, Germany). With the help of a BioSpec-nano micro-volume UV–Vis spectrophotometer (Shimadzu Co., Kyoto, Japan), the purity and concentration of the total RNA were assessed.

### 2.4. Microarray Hybridization and Data Analysis

The miRNA profile was determined by the Affymetrix Microarray system using the GeneChip miRNA 4.0 Array (version 4.0.0.1567G) (Affymetrix, Santa Clara, CA, USA). The miRNA microarray contains 30,434 mature miRNAs from all organisms, miRBase set Release 20.0. Following the manufacturer’s instructions, 500 ng of total RNA was labeled using an Affymetrix FlashTag Biotin HSR RNA Labeling Kit. Following hybridization, the chip arrays were washed and stained using AGCC Fluidics Control Software (version 1.4.1) on a Fluidics Station 450. After that, a high-resolution Affymetrix GeneChip Scanner 3000 was used to scan the array’s fluorescence. The Affymetrix GeneChip Command Console program produced probe cell intensity files (*CEL files), which were imported into probe-level summary files (*CHP files) for data extraction. Software called the Expression Console and Transcriptome Analysis Console (version 4.0.2) were used to analyze these *CHP files. After image analysis, the data were normalized. The data were compared using tests, including the t-test, analysis of variance (ANOVA), and significance analysis of microarray after log_2_ translation of the normalized signal intensities of the arrays. The identification of miRNA-targeted genes was performed using TargetScan [31] (http://www.targetscan.org; accessed on 8 January 2024), with the selection criteria of cumulative weighted value ≤ −0.75 for Targetscan. Genes targeted by multiple miRNAs were assembled from individual miRNA target lists. ConsensusPathDB [32] (http://cpdb.molgen.mpg.de/; accessed on 10 January 2024) was used to perform gene ontology (GO) annotation and enrichment analysis from three ontologies (molecular function, biological process, and cellular component) on the frequently identified target genes. The threshold for significance was set at *p* < 0.01 for both of these analyses. The predicted target gene lists were examined using the default collections of the KEGG, Reactome, and BioCarta (http://cgap.nci.nih.gov/Pathways/BioCarta_Pathways; accessed on 11 January 2024) databases.

## 3. Results

The expression profiles of miRNAs in the fetal *longissimus dorsi* muscle that were collected from ewes fed with the SP, HP, and LP were compared using the microarray analysis. As presented in Table 1, 12 different miRNAs were differentially expressed (>2-fold, *p* < 0.01, FDR < 0.05) in the different diet groups, as per the Affymetrix GeneChip miRNA 4.0 Array. The chromosomal locations and mature miRNA sequences of these 12 prominent miRNAs identified using the Transcriptome Analysis Console commercial software (version 4.0.2) are shown in Table 1.

We identified two significantly up-regulated miRNAs in the HP group compared to the SP group (Table 1). The two miRNAs were oar-miR-3957-5p and oar-miR-329a-3p. The fold change of those miRNAs was 4.58 and 2.96, respectively. The oar-miR-3957-5p miRNA was also up-regulated in the SP group compared to the HP group, with a fold change of 4.25. One of the remarkable results of our miRNAome analysis was the identification of nine miRNAs on OAR18, in which they clustered together in a tight locus (Appendix A). The oar-miR-200c miRNA showed up-regulation in the SP group compared to the LP group. The differentially expressed miRNAs were grouped using a hierarchical clustering technique, and the outcomes demonstrated that the various diet groups could be readily distinguished by the hierarchical grouping of all covered ovine miRNAs (Appendix A).

Next, we analyzed the target genes that may be regulated by these 12 miRNAs in two layers. In the first layer, we identified the genes targeted by multiple miRNAs (Appendix A). The in silico analyses identified that 32 genes were found to be targeted by two miRNAs, and no mRNA was found to be targeted by three or more miRNAs (Appendix A). In the second layer, a total of 65 genes were targeted by at least one of twelve miRNAs (Appendix A). Hypergeometric distribution testing was used to examine the gene ontology (GO) of their anticipated target genes to investigate the roles of differentially regulated miRNAs among three distinct diet groups. The GO terms for the genes that were targeted by two miRNAs were identified as being related to 19 biological processes (BP) and six molecular functions (MF) (Appendix A). The GO terms for the genes that were targeted by single miRNA were identified as being related to 20 biological processes, which were involved in the creation of five molecular functions and were expected to take part in three cellular components (CC) (Appendix A).

The GO functional enrichment and pathway enrichment analyses were performed on 32 genes targeted by multiple miRNAs and separately on 65 genes targeted by at least one miRNA (Appendix A). The GO terms showed that genes targeted by multiple miRNAs were mainly involved in BP lymphocyte chemotaxis, monocyte chemotaxis, the regulation of the nitrogen compound metabolic process, granulocyte chemotaxis, and other categories that control the immune system (Appendix A). In terms of the MF, targeted mRNAs were involved in cytokine activity, cytokine receptor binding, and protein kinase regulator activity (Appendix A). We conducted a pathway enrichment study using public pathway databases to better understand the roles of these projected target genes, and 31 pathways were found (Appendix A). The pathway analysis showed that genes targeted by multiple miRNAs were significantly enriched in chemokine receptors bind chemokines, rheumatoid arthritis-*Homo sapiens* (human), viral protein interaction with cytokine and cytokine receptor-*Homo sapiens* (human), and the toll-like receptor signaling pathway-*Homo sapiens* (human) (Appendix A).

For the GO functional enrichment analysis for the genes targeted by single mRNAs in terms of the BP, the primary categories for the functions were the cellular macromolecule biosynthetic process, macromolecule biosynthetic process, regulation of the biosynthetic process, and the cellular nitrogen compound biosynthetic process cell (Appendix A). In terms of the MF, targeted mRNAs were involved in DNA binding, regulatory region nucleic acid binding, and metal ion binding (Appendix A). In terms of the CC, targeted mRNAs were involved in the nuclear lumen, nucleus, and nucleoplasm (Appendix A). The pathway analysis showed that genes targeted by single miRNAs were significantly enriched in rheumatoid arthritis-*Homo sapiens* (human), transferrin endocytosis and recycling, oncogene-induced senescence, and G1 to S cell-cycle control (Appendix A).

## 4. Discussion

Proteins are vital for the structure, function, and regulation of the body’s tissues and organs in all living organisms. In addition, they are the most expensive and often most limiting ingredients in livestock feed formulation [33]. Excess protein intake can increase nitrogen excretion, which can lead to unfavorable nitrate leakage into groundwater and emissions of ammonia into the atmosphere. [34]. Despite the importance of protein for all living organisms, ruminants use dietary protein for production rather inefficiently (as defined by g N in product/g N intake) [35]. This creates an opportunity to utilize technologies such as dynamically altering nutrient intakes via mid-infrared spectroscopy of milk composition [34,36]. Additionally, feedomics may assist in modifying nutrition for monetary and long-term gains. The key components of the feedomics study are nutrigenomics, nutrigenetics, and nutritional epigenetics, which aid in establishing a link between genetic variation and nutrient-driven epigenetic changes that are proposed as the main barrier to meeting nutritional demands [37]. Maternal nutrition is a major intrauterine environmental factor that alters the expression of the fetal genome in ways that may have lifelong consequences. Such fetal programming has been shown to be regulated by epigenetic modifications, including DNA methylation, histone modifications, as well as miRNAs [22].

In addition, we have identified that miR-3957-5p was up-regulated in both the LP and SP groups vs. the HP group (Table 1). Pokharel et al. [38] detected miR-3957-5p in the tissue biopsies of the corpus luteum and endometrium of pregnant ewes. Prior work demonstrated an abundance of miRNAs in packed form as extracellular vesicles to maintain feto–maternal communication in multiple mammals [39,40,41]. In the current study, together with miR-3957-5p, other miRNAs revealed a large cluster of miRNAs on OAR18 (Table 1), which may co-regulate different biological processes [42]. This is consistent with Pokharel et al. [38], who also detected similar miRNAs in the same clustered locus. This miRNA cluster is highly conserved among placental mammals and is known to be regulated by a maternally imprinted *DLK1-DIO3* region, which has been associated with severe muscle disease, which is caused by the impaired expression of dystrophin called Duchenne muscular dystrophy [43,44,45]. Additionally, two growth QTLs associated with sheep body weight [46] and average daily gain [47] were found to coincide with the miRNA cluster on OAR18. In the same experimental design subjected to the current study, Sohel et al. [18] showed that a high-protein diet consumed by the mother predominantly changed the expression of mRNAs involved in myogenesis and immunity in the developing fetus’s skeletal muscle. Both Shandilya et al. [48] and Sharma et al. [49,50] showed that sheep exposed to a bacterial lipopolysaccharide challenge exhibited differentially regulated miRNAs, such as miR-3957-5p, miR-329a-3p, and miR-379-5p (Table 1) which regulate biological processes such as stress and immune responses.

A critical layer of analysis involves miRNA targeting the same genes (Appendix A). This is an established part of miRNA biology and a useful way to highlight important genes and pathways [51,52]. Multiple miRNAs regulating the same target gene are naturally cooperative since miRNAs act to inhibit targets; this heterotypic regulation is believed to offer extra regulatory specificity [53]. And especially, there is an ongoing debate over whether clustered miRNAs with different seed sequences co-evolved due to shared functions [53]. In the current study, two members of the miRNA cluster, which are oar-miR-3957-5p and oar-miR-432 (Appendix A; Appendix A), are cooperatively targeting the calcium/calmodulin-dependent protein kinase IG (*CAMK1G*) and ras protein-specific guanine nucleotide releasing factor 1 (*RASGRF1*) genes. Calcium/calmodulin-dependent protein kinases are known to be key regulators of calcium signaling in health and disease [54], and it has been observed that T cells and B cells produce less IFN-γ when *CaMKIV* is inhibited or silent [55]. Furthermore, the experiments showed that in myoblasts and myotubes treated with IFN-γ, calcium/calmodulin-dependent protein kinase IV specifically suppresses the production of co-stimulatory molecules and pro-inflammatory cytokines/chemokines [56]. In terms of *RASGRF1*, it is an intrinsic key mediator for brain-derived neurotrophic factor (BDNF)-induced R-Ras activation and R-Ras-mediated axonal morphological regulation [57]. One of the nerve growth factor family members, BDNF, is primarily produced by the brain, and its main role involves synaptic modulation, neurogenesis, neuron survival, immune regulation, myocardial contraction, and angiogenesis in the brain [58]. On the other hand, previous experiments have shown that BDNF has a significant regulatory role in myogenic differentiation in skeletal muscle, which acts as a retrograde survival factor for motor neurons in the neuromuscular system [59,60]. Additionally, on the same miRNA cluster, oar-miR-379-5p and oar-miR-134-5p were found to be cooperatively targeting Glutamyl-TRNA Amidotransferase Subunit C (*GATC*) (Appendix A). *GATC* was associated with mitochondrial respiratory dysfunction, which causes skeletal muscle atrophy in a state of uncontrolled inflammation and oxidative stress [61]. Multiple target analyses showed a complex network of mutual interactions between let-7c, miRNAs on various chromosomes, and mRNAs (Appendix A). In the literature, the role of Let-7c and other Let-7 family members in myogenesis [62] and ovine fetal skeletal muscle development related to maternal obesity has been shown [63].

A second outcome involves the immune genes. Chemokine ligands (CCLs), *CCL3*, *CCL3L1*, and *CCL3L3* are all targeted by both oar-miR-432 and let-7c (Appendix A). *CCL3* is critical for trophoblast invasion and the maintenance of pregnancy [64,65]. While others have studied dietary impacts on sheep, to our knowledge, this is the first demonstration that maternal diet protein intakes in the context of isoenergetic dietary formulations could impact *CCL3* and the maintenance of pregnancy. In addition, *IL7* is targeted by both oar-miR-432 and oar-miR-200c (Appendix A). This is particularly striking because *IL7* influences the development of the offspring’s T cell immunity [66,67]. These examples highlight the importance of maternal protein intake on multiple critical outcomes for fetal health and lifetime well-being.

Rheumatoid arthritis-*Homo sapiens* (human) pathways were identified in different protein intake groups among both layers of the target prediction (Appendix A). Rheumatoid arthritis (RA) is a chronic, inflammatory, systemic autoimmune disease that causes skeletal muscle loss and weakness in RA individuals [68,69]. Similarly, transferrin endocytosis and recycling pathways are associated with iron homeostasis, a key regulator of innate and adaptive immunity, and play a decisive role in inflammatory processes [70]. These findings were consistent with the findings of the transcriptome analysis [28], which revealed that many detected genes are involved in the pathways related to the immune system and diseases.

## 5. Conclusions

The findings of our research suggested that maternal diets of different protein levels affected fetal miRNAome, which epigenetically affects the skeletal muscle transcriptome in vivo. The discovery of multiple miRNAs that target mRNAs is a significant development that may enable us to further understand the network that connects miRNAs and their targets. Since the maternal HP diet specifically changed the expression of the miRNAs involved in a number of pathways and biological processes that are unique to disease and immunity, these results might have provided insight into the reversible nature of epigenetic modifications. New studies are warranted for the expression levels of already detected fetal miRNAs in different environmental conditions, breeds, etc., in the offspring.

## Figures and Tables

**Table 1 animals-14-01594-t001:** The list of differentially expressed miRNAs with a fold change ≥ 2 and FDR = 0 in fetal skeletal muscles derived from HP, LP, and SP groups.

Transcript ID (Array Design)	miRBase ID	Fold Change	*p*-Value	Style	Mature Sequence	OAR
LP vs. HP						
oar-miR-3957-5p	MIMAT0019325	4.58	0.04	up	cucggagaguggagcugugggugu	18
oar-miR-329a-3p	MIMAT0019266	2.96	0.04	up	aacacaccugguuaaccuuuuu	18
SP vs. HP						
oar-miR-3957-5p	MIMAT0019325	4.25	0.005	up	cucggagaguggagcugugggugu	18
oar-miR-432	MIMAT0001416	2.47	0.007	up	ucuuggaguaggucauugggugg	18
oar-miR-200c	MIMAT0030044	2.24	0.009	up	uaauacugccggguaaugaugg	3
oar-miR-362	MIMAT0030060	2.24	0.01	up	aauccuuggaaccuaggugugagu	X
oar-miR-409-3p	MIMAT0019328	2.2	0.01	up	cgaauguugcucggugaaccccu	18
oar-let-7c	MIMAT0014964	2.18	0.01	up	ugagguaguagguuguaugguu	1
oar-miR-493-3p	MIMAT0019238	2.17	0.01	up	ugaaggucuacugugugccagg	18
oar-miR-181a	MIMAT0014973	2.12	0.02	up	aacauucaacgcugucggugagu	12
oar-miR-379-5p	MIMAT0019247	2.11	0.02	up	ugguagacuauggaacguaggc	18
oar-miR-134-5p	MIMAT0019308	2.1	0.03	up	ugugacugguugaccagaggg	18
oar-miR-127	MIMAT0001415	2.04	0.04	up	aucggauccgucugagcuuggcu	18
SP vs. LP						
oar-miR-200c	MIMAT0030044	2.45	0.01	up	uaauacugccggguaaugaugg	3

## Data Availability

All data generated and analyzed in this study are presented in the figures/tables and Appendix A.

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
