# Peer review of "Effects of Low and High Maternal Protein Intake on Fetal Skeletal Muscle miRNAome in Sheep"

_animals, 2024, doi:10.3390/ani14111594_

Round 1

Reviewer 1 Report

Comments and Suggestions for Authors

REVIEW

Introduction to the subject of research

Despite continuous improvements in herd management and intensive research into mammalian nutritional requirements using modern analytical techniques, IUGR continues to be a major problem in animal husbandry due to incomplete knowledge of the impact of nutrition on fetal growth regulation mechanisms (Wu et al. , 2004a; McMillen and Robinson, 2005; Murphy et al. , 2006). Increasing the number of foetuses in the uterus without increasing its capacity results in relative placental insufficiency and low birth weight of the young (Gootwine et al. , 2006). In the case of livestock, the primary cause is an incorrectly balanced diet. Interestingly, both low-protein and high-protein diets are the cause of intrauterine growth inhibition syndrome (Mickiewicz et al. , 2012). One of the main causes of this phenomenon is the delay in blastocyst development caused by the increased amount of toxic protein breakdown products circulating in the maternal body (Meza-Herrera et al. , 2010). On a high-protein diet, there is also an increase in the energy costs of urine production and hepatic gluconeogenesis, which results in a reduced supply of energy components to the fetus. Studies in pregnant rodents and sheep have shown that high protein intake and increased ammonia levels (up to 300% higher than controls) reduce the number of developing blastocysts, impair embryo metabolism and lead to foetal growth disorders (Lane and Gardner, 2003; Gardner et al. , 2004). On a high-protein diet, the amino acid profile of the pregnant mother changes, the concentration of threonine, glutamine, glycine, alanine and serine decreases, which affects the slowing of fetal tissue growth (Rehfeldt et al. , 2011). On a high-protein diet, the amino acid profile of the pregnant mother changes, the concentration of threonine, glutamine, glycine, alanine and serine decreases, which affects the slowing of fetal tissue growth (Rehfeldt et al. , 2011). In mice, the effects of a high-protein diet on the GH/JAK/STAT/IGF pathway were observed, resulting in decreased placental growth hormone levels and consequently decreased foetal weight gain (Vanselow et al. , 2011). Changes in lipid metabolism result in an increase in the level of annexin IV, the protein responsible for the differentiation of adipocytes, and an increase in adipose tissue (Sarr et al. , 2011). This translates into an increased predisposition to fattening adults. An indirect effect of consuming excess protein during pregnancy, but extremely important in the adult life of individuals born with intrauterine growth retardation syndrome, is the changes occurring within the circulatory system, which leads to the development of cardiovascular diseases. The reduced protein content of the pregnant female diet, as with the high protein diet, results in a decrease in birth weight (Davis et al. , 1997). The effect of reducing dietary protein intake is to reduce the pool of amino acids in maternal blood plasma and consequently in fetal blood plasma and uterine fluid. This, in turn, translates into inhibition of fetal development (Wu et al. , 1998).

The aim of this study was to investigate the effect of different levels of protein in the diet of mid- and late-pregnancy sheep on the total expression of foetal skeletal miRNA in sheep.

Title

The title of the article is adequate to the presented research results.

Abstract

The abstract of the work is correctly written contains the most important results of the experiment

Introduction

The chapter is correctly written, but there is no elaboration of the problem concerning intrauterine growth restriction IUGR

Methodology

The research material and the analytical and statistical methods used were appropriate to the scope of the present studies.

Measurements of sheep body weight at the beginning and end of the experiment were not included in the work. Missing from the description of fetal body weight measurements.

Results

The results were collected in the form of 5 table with description.

There was no very relevant information about the mothers’ body weight at the beginning and end of the experiment. No foetal body weight was reported.

Discussion

The discussion is conducted properly but it does not address the most important problem with maternal nutrition during pregnancy, namely inside the uterine restriction of fetal growth IUGR. It is not enough to base the assumptions of the work solely on genetic markers.

Author Response

Reviwer 1:

Comment 1: introduction to the subject of research

Despite continuous improvements in herd management and intensive research into mammalian nutritional requirements using modern analytical techniques, IUGR continues to be a major problem in animal husbandry due to incomplete knowledge of the impact of nutrition on fetal growth regulation mechanisms (Wu et al., 2004a; McMillen and Robinson, 2005; Murphy et al., 2006). Increasing the number of foetuses in the uterus without increasing its capacity results in relative placental insufficiency and low birth weight of the young (Gootwine et al., 2006). In the case of livestock, the primary cause is an incorrectly balanced diet. Interestingly, both low-protein and high-protein diets are the cause of intrauterine growth inhibition syndrome (Mickiewicz et al., 2012). One of the main causes of this phenomenon is the delay in blastocyst development caused by the increased amount of toxic protein breakdown products circulating in the maternal body (Meza-Herrera et al., 2010). On a high-protein diet, there is also an increase in the energy costs of urine production and hepatic gluconeogenesis, which results in a reduced supply of energy components to the fetus. Studies in pregnant rodents and sheep have shown that high protein intake and increased ammonia levels (up to 300% higher than controls) reduce the number of developing blastocysts, impair embryo metabolism and lead to foetal growth disorders (Lane and Gardner, 2003; Gardner et al., 2004). On a high-protein diet, the amino acid profile of the pregnant mother changes, the concentration of threonine, glutamine, glycine, alanine and serine decreases, which affects the slowing of fetal tissue growth (Rehfeldt et al., 2011). On a high-protein diet, the amino acid profile of the pregnant mother changes, the concentration of threonine, glutamine, glycine, alanine and serine decreases, which affects the slowing of fetal tissue growth (Rehfeldt et al., 2011). In mice, the effects of a high-protein diet on the GH/JAK/STAT/IGF pathway were observed, resulting in decreased placental growth hormone levels and consequently decreased foetal weight gain (Vanselow et al., 2011). Changes in lipid metabolism result in an increase in the level of annexin IV, the protein responsible for the differentiation of adipocytes, and an increase in adipose tissue (Sarr et al., 2011). This translates into an increased predisposition to fattening adults. An indirect effect of consuming excess protein during pregnancy, but extremely important in the adult life of individuals born with intrauterine growth retardation syndrome, is the changes occurring within the circulatory system, which leads to the development of cardiovascular diseases. The reduced protein content of the pregnant female diet, as with the high protein diet, results in a decrease in birth weight (Davis et al., 1997). The effect of reducing dietary protein intake is to reduce the pool of amino acids in maternal blood plasma and consequently in fetal blood plasma and uterine fluid. This, in turn, translates into inhibition of fetal development (Wu et al., 1998).

The aim of this study was to investigate the effect of different levels of protein in the diet of mid- and late-pregnancy sheep on the total expression of foetal skeletal miRNA in sheep.

 Title

The title of the article is adequate to the presented research results.

 Abstract

The abstract of the work is correctly written contains the most important results of the experiment

Response 1: Thanks for careful read.

Comment 2: Introduction

The chapter is correctly written, but there is no elaboration of the problem concerning intrauterine growth restriction IUGR

Response 2: A paragraph that tackle relation between maternal protein intake and intrauterine growth restriction was added in the introcduction section and references were also added in the reference list.

Comment 3: Methodology

The research material and the analytical and statistical methods used were appropriate to the scope of the present studies.

Measurements of sheep body weight at the beginning and end of the experiment were not included in the work. Missing from the description of fetal body weight measurements.

Response 3: Body weights of ewe’s and fetuses were given in the Sohel et al. (2020) as a supplementary data except with fetus weights. The data was given as supplementary Table in the revised manuscript.

Sohel, M.M.H., Akyuz, B., Konca, Y. et al. Differential protein input in the maternal diet alters the skeletal muscle transcriptome in fetal sheep. Mamm Genome 31, 309–324 (2020).

Table S1: Ewe and fetus specifications

Parameters

Standard Protein

High protein

Low protein

SEM

P value

Number

4

4

4

Breed

Akkaraman

Akkaraman

Akkaraman

-

-

Age

2 years

2 years

2 years

-

-

Live Weight at the beginning (kg)

58.40

57.00

57.00

2.004

0.954

30. day live weight (kg)

58.22

57.94

57.96

2.089

0.998

105. day live weight (kg)

75.50

74.00

62.50

4.631

0.568

Fetal weight (kg)

1.09

1.11

1.13

0.22

0.587

Comment 4: Results

The results were collected in the form of 5 table with description.

There was no very relevant information about the mothers’ body weight at the beginning and end of the experiment. No foetal body weight was reported.

Response 4: Body weights of ewe’s and fetuses were given in the Sohel et al. (2020) as a supplementary data except with fetus weights. The data was given as supplementary Table in the revised manuscript.

Sohel, M.M.H., Akyuz, B., Konca, Y. et al. Differential protein input in the maternal diet alters the skeletal muscle transcriptome in fetal sheep. Mamm Genome 31, 309–324 (2020).

Comment 5: Discussion

The discussion is conducted properly but it does not address the most important problem with maternal nutrition during pregnancy, namely inside the uterine restriction of fetal growth IUGR. It is not enough to base the assumptions of the work solely on genetic markers.

Response 5: Our hypothesis was whether different levels of maternal protein intake has effect on the foetal skeletal muscle miRNA expression. Although, we recorded foetal weights, no statistical differences were found. (Please see table above).

Reviewer 2 Report

Comments and Suggestions for Authors

The authors show that dietary protein intake during pregnancy affects fetal skeletal muscle epigenetics via miRNA expression.

Main question addressed by the study.

The authors address potential effects of nutrition in pregnant small ruminants.

Novelty.

The main finding of the study is that most genes targeted by identified miRNAs are involved in immune and muscle disease pathways. This must be clearly underlined in the abstract and in the conclusions of the manuscript.

This is a gap in the literature and the authors should exploit this further within their text.

The authors should make clear that this is a finding additional to the published information in other relevant papers.

Methodologies.

The study lacks any analysis. First, please confirm the normal distribution of the data obtained in the study.

Controls. Do the authors have comparative findings with other breeds of sheep? Do they have control animals with various types of management: extensive versus intensive and grazing versus housed animals? The authors should carry out the additional experiments to produce control findings and then they should add the new results in the revised manuscript.

These further controls will greatly benefit the study and the manuscript.

References.

All the references are appropriate. No need to add or remove references.

Comments on tables and figures.

The presentation of the results in this format makes difficult the understanding by readers. I suggest to move the current tables in the Appendix, as they are summarized well in the main text.

Also, I suggest to include graphs with the salient results of the study. At least 4 to 5 graphs must be constructed and added to enhance the technical quality of the manuscript. Adding graphs in the revised text will help future readers to grasp better the findings and will help the manuscript to receive citations.

Discussion and Conclusions.

The problem in these findings is the lack of extending the findings to other breeds of sheep, as the results, as they are now, are rather of limited interest. I suggested to extend the experiment by adding more breeds of sheep, Lacaune from France, Assaf from Spain and Suffolk from United Kingdom. This will make the manuscript strong and interesting and of international interest as well.

The Conclusions must reflect the limited significance of the findings and therefore they must be rewritten to be consistent with the findings of the study.

Overall.

The recommendation is: please carry out more experiments, prepare a fully revised manuscript and resubmit for further evaluation.

Author Response

Reviewer 2:

Comment 1: The authors show that dietary protein intake during pregnancy affects fetal skeletal muscle epigenetics via miRNA expression.

Main question addressed by the study.

The authors address potential effects of nutrition in pregnant small ruminants.

Novelty.

The main finding of the study is that most genes targeted by identified miRNAs are involved in immune and muscle disease pathways. This must be clearly underlined in the abstract and in the conclusions of the manuscript.

This is a gap in the literature and the authors should exploit this further within their text.

Response 1: One of the main outcomes of our study is the enrichment of immune related genes that are targeted by multiple or single miRNAs. Pathways and gene ontology terms associated with those genes were given different sections such as absract (L36-37), results (Tables 2-5), discussion (L350-368) and conclusion (L380-384) of the manuscript. 

Comment 2: The authors should make clear that this is a finding additional to the published information in other relevant papers.

Response 2: In literature, mainly association between maternal diet modifications and fetal phenotypes such as weight, organ weights and morphological variations were investigated. However, our literature search showed us association between maternal protein intake during gestation and fetal miRNA expressions was generally insufficient in mammalian species and there was no study specific to sheep. Therefore, our study was designed to investigate association between maternal protein intake during gestation and fetal miRNA expressions.    

Comment 3: Methodologies.

The study lacks any analysis. First, please confirm the normal distribution of the data obtained in the study.

Response 3: miRNA expression analysis was done by providing service from Affymetrix dealer. For expression analysis Transcriptome Analysis Console (TAC) (Thermo Fisher) was used. We followed analysis pipeline provided by company. (https://www.thermofisher.com/tr/en/home/life-science/microarray-analysis/microarray-analysis-instruments-software-services/microarray-analysis-software/affymetrix-transcriptome-analysis-console-software.html).

Comment 4 : Controls. Do the authors have comparative findings with other breeds of sheep? Do they have control animals with various types of management: extensive versus intensive and grazing versus housed animals? The authors should carry out the additional experiments to produce control findings and then they should add the new results in the revised manuscript.

These further controls will greatly benefit the study and the manuscript.

Response 4: Role of epigenome including miRNAs in fetal development is known as stated in the introduction. On the other hand, given the role of miRNAs in fetal development and the sensitivity of pregnant dams to environmental factors such as diet, we hypothesize that maternal nutrition during pregnancy alters expression levels of miRNAs in fetus via epigenetic modifications, which in turn would lead to changes in fetal and adult programing. In this study, we report the impact of maternal dietary protein intake during gestation on expression of whole genome miRNAs in fetal skelatal muscle tissue in sheep model. In the literature, it has been reported that sheep is a suitable large animal model for genetics, neurology, hearing, immunology, etc. studies. Literature that used sheep as model experimental animal has been given below.

  • Banstola A, Reynolds JNJ. The Sheep as a Large Animal Model for the Investigation and Treatment of Human Disorders. Biology (Basel). 2022 Aug 23;11(9):1251. doi: 10.3390/biology11091251.
  • Li M, Lu Y, Gao Z, Yue D, Hong J, Wu J, Xi D, Deng W, Chong Y. Pan-Omics in Sheep: Unveiling Genetic Landscapes. Animals (Basel). 2024 Jan 15;14(2):273. doi: 10.3390/ani14020273. 
  • Miyasaka, Masayuki, and Zdenek Trnka. "Sheep as an experimental model for immunology: immunological techniques in vitro and in vivo." Immunological methods. Academic Press, 1985. 403-423.
  • Lue, Po-Yi, et al. "Sheep as a large animal model for hearing research: comparison to common laboratory animals and humans." Laboratory Animal Research 39.1 (2023): 31.

For this reason, sheep was preferred as the experimental animal in our study. We aimed to investigate how maternal protein intake (low, standard, and high) during pregnancy, affects miRNA expression in fetal skeletal muscle. For the reasons stated above, sheep were chosen as the model animal to achieve this purpose. In addition, it was thought that the study results regardless of species could serve as an example for other mammal species too. Therefore, our study does not include investigation of the effects of species, especially breeds, and different sheep husbandry systems.

In literature search, no study similar to our study design was found in sheep and in other mammalian livestock. Studies investigating the fetal nutrigenomic effects of maternal nutrition during pregnancy are given in the table below. As can be seen from the table, breed was not considered in maternal nutrigenomic studies.

Taken to gether, standart protein group was considered as control group compared low and high protein groups in the current study.

Breed

Experiment

Effect

Reference

Polypay ewes

Gene Expression and DNA Methylation

Alfalfa haylage, corn, or dried corn distiller's grains

Lan X, Cretney EC, Kropp J, Khateeb K, Berg MA, Peñagaricano F, Magness R, Radunz AE, Khatib H. Maternal Diet during Pregnancy Induces Gene Expression and DNA Methylation Changes in Fetal Tissues in Sheep. Front Genet. 2013 Apr 5;4:49. doi: 10.3389/fgene.2013.00049.

Polypay ewes

Methylomic and transcriptomic

Alfalfa haylage, corn, or dried corn distiller's grains

Namous H, Peñagaricano F, Del Corvo M, Capra E, Thomas DL, Stella A, Williams JL, Marsan PA, Khatib H. Integrative analysis of methylomic and transcriptomic data in fetal sheep muscle tissues in response to maternal diet during pregnancy. BMC Genomics. 2018 Feb 6;19(1):123. doi: 10.1186/s12864-018-4509-0. 

Polypay ewes

Gene expression changes in fetal muscle and adipose tissues

Alfalfa haylage, corn, or dried corn distiller's grains

Peñagaricano F, Wang X, Rosa GJ, Radunz AE, Khatib H. Maternal nutrition induces gene expression changes in fetal muscle and adipose tissues in sheep. BMC Genomics. 2014 Nov 28;15(1):1034. doi: 10.1186/1471-2164-15-1034.

Angus cross cows

Gene expression analysis

Low starch (LS; haylage) and high starch (HS; corn silage).

Wang X, Lan X, Radunz AE, Khatib H. Maternal nutrition during pregnancy is associated with differential expression of imprinted genes and DNA methyltranfereases in muscle of beef cattle offspring. J Anim Sci. 2015 Jan;93(1):35-40. doi: 10.2527/jas.2014-8148. 

Columbia/Rambouillet cross ewes

Expression of enzymes

Diet of 100% (Control) or 150% (Obese) of NRC

Long NM, Rule DC, Zhu MJ, Nathanielsz PW, Ford SP. Maternal obesity upregulates fatty acid and glucose transporters and increases expression of enzymes mediating fatty acid biosynthesis in fetal adipose tissue depots. J Anim Sci. 2012 Jul;90(7):2201-10. doi: 10.2527/jas.2011-4343. 

German Landrace sows

Gene expression and methylation

Dietary protein restriction

Altmann S, Murani E, Schwerin M, Metges CC, Wimmers K, Ponsuksili S. Maternal dietary protein restriction and excess affects offspring gene expression and methylation of non-SMC subunits of condensin I in liver and skeletal muscle. Epigenetics. 2012 Mar;7(3):239-52. doi: 10.4161/epi.7.3.19183.

Western White-faced ewes

Transcriptome (mRNA)

Control (100% NRC)-, restricted (60% NRC)-, or overfed (140% NRC)

Gauvin MC, Pillai SM, Reed SA, Stevens JR, Hoffman ML, Jones AK, Zinn SA, Govoni KE. Poor maternal nutrition during gestation in sheep alters prenatal muscle growth and development in offspring. J Anim Sci. 2020 Jan 1;98(1): skz388. doi: 10.1093/jas/skz388.

Comment 5: References.

All the references are appropriate. No need to add or remove references.

Response 5: Thanks for careful check.

Comment 6: Comments on tables and figures.

The presentation of the results in this format makes difficult the understanding by readers. I suggest to move the current tables in the Appendix, as they are summarized well in the main text.

Response 6: Tables 2-5 were given as Appendix and modified in the revised mansucript (headings were highligted with yellow color).

Comment 7:  Also, I suggest to include graphs with the salient results of the study. At least 4 to 5 graphs must be constructed and added to enhance the technical quality of the manuscript. Adding graphs in the revised text will help future readers to grasp better the findings and will help the manuscript to receive citations.

Response 8: Gene ontology terms of 65 genes with ConsensusPathDB software web tool is providing following graph. Since graph visulizations seemed complicated, we prefered to give the results as table.  

Comment 8: Discussion and Conclusions.

The problem in these findings is the lack of extending the findings to other breeds of sheep, as the results, as they are now, are rather of limited interest. I suggested to extend the experiment by adding more breeds of sheep, Lacaune from France, Assaf from Spain and Suffolk from United Kingdom. This will make the manuscript strong and interesting and of international interest as well.

Response 8: We aimed to investigate how maternal protein intake (low, standard, and high) during pregnancy, affects miRNA expression in fetal skeletal muscle. Sheep were chosen as the model animal to achieve this purpose in the current study. Therefore, our study does not include investigation of the effects of different breeds.

Comment 9: The Conclusions must reflect the limited significance of the findings and therefore they must be rewritten to be consistent with the findings of the study.

Response 9: Thank you for the valuable opinion of the esteemed reviewers.

However, it has been shown that protein supplementation of expectant mothers during pregnancy causes some phenotypic changes in the offspring. This study aimed to investigate which miRNA expression related to quantitative characteristics in offspring muscle tissue is affected by protein restriction or increased protein intake during pregnancy. Therefore, the conclusion part is prepared with the results of our experiment to test the study hypothesis.

As we mentioned before, sheep is the experimental animal we chose to test our hypothesis due to its advantages. We think that our results may be of interest to all mammal species, including humans, regardless of breed. Therefore, race and upbringing are not important factors for this study.

Overall.

Comment 10: The recommendation is: please carry out more experiments, prepare a fully revised manuscript and resubmit for further evaluation.

Response 10: Above we tried to explain the hypothesis and aim of the study what we have performed. What the reviewer wants is the subject of another study independent of ours. As a team, we would like to thank you for your valuable opinions.

Reviewer 3 Report

Comments and Suggestions for Authors

The objective of this study was to investigate the impact of different maternal dietary protein levels during mid-to-late gestation on the total miRNA expression of fetal skeletal muscle in sheep.

-The authors should make clear the main and the secondary questions addressed by their work. At the moment, this is not clear to readers, so please make appropriate changes.

-There are similar studies in the international literature. The authors must underline the gaps in the international literature that could be filled through the results of their work. Which of this novel knowledge is relevant for improving management of sheep in the future?

Moreover, it this new knowledge applicable to goats for which species data are missing? Can the authors explain if there are benefits for that animal species as well?

-In describing the potential novelty in this study, please explain clearly the advancements achieved by this study over previously published papers.

-The controls in the study are not clearly presented. Please add a new subsection to describe how you dealt with controls. Please note that explanations about positive and negative controls should be presented. Please describe accurately the controls for all stages of the study (animals, laboratory testing, material used, environmental conditions etc.).

-With regard to twin fetuses, please present the results in a separate group and make comparisons with the group of ewes with one fetus.

-Comments on tables and figures. The tables are long and difficult to read. No figures have been included. Please move some tables to supplementary material. Please add 3 to 5 figures in order to make the results more easily understood. Please include photographs from the experimental work (e.g., gels or microarrays).

-The Discussion is a bit shallow and does not fully address all the results obtained. Please rewrite by extending it and by making reference to the specific findings for each stage of the study.

Please include a paragraph about commercialisation of the findings. Is there a patent pending? You need to mention how this will be financed in clinical practice.

-Conclusions. Please avoid to introduce new ideas in Conclusions. Some of the Conclusions are a bit optimistic and not fully justified by findings. These need to be toned down.

What are the practical advantages of the findings for the average veterinarian in practice? How these findings can improve cash flow and larger clientele? Please add these in the revised version.

Overall. The manuscript needs significant improvement before possible acceptance. Re-evaluation is necessary after implementing the requested changes.

Author Response

Reviewer 3:

The objective of this study was to investigate the impact of different maternal dietary protein levels during mid-to-late gestation on the total miRNA expression of fetal skeletal muscle in sheep.

Comment 1: -The authors should make clear the main and the secondary questions addressed by their work. At the moment, this is not clear to readers, so please make appropriate changes.

Response 1:  The sentence was modified:

From: The objective of this study was to investigate the impact of different maternal dietary pro-tein levels during mid-to-late gestation on the total miRNA expression of fetal skeletal muscle in sheep.

To: The objective of this study was thus to determine the consequences of different maternal dietary protein uptake throughout gestation on sheep fetuses as follows: 1) on the expres-sion of whole genome skeletal muscle miRNAs; 2) biological pathways and gene ontology related with the miRNAome; and 3) understanding the connections between miRNAs and their targets.

Comment 2:-There are similar studies in the international literature. The authors must underline the gaps in the international literature that could be filled through the results of their work. Which of this novel knowledge is relevant for improving management of sheep in the future?

Response 2: In literature search, no study similar to our study design was found in sheep and in other mammalian livestock. Studies investigating the fetal nutrigenomic effects of maternal nutrition during pregnancy are given in the table below. As can be seen from the table, breed was not considered in maternal nutrigenomic studies.

Taken to gether, standart protein group was considered as control group compared low and high protein groups in the current study.

Breed

Experiment

Effect

Reference

Polypay ewes

Gene Expression and DNA Methylation

Alfalfa haylage, corn, or dried corn distiller's grains

Lan X, Cretney EC, Kropp J, Khateeb K, Berg MA, Peñagaricano F, Magness R, Radunz AE, Khatib H. Maternal Diet during Pregnancy Induces Gene Expression and DNA Methylation Changes in Fetal Tissues in Sheep. Front Genet. 2013 Apr 5;4:49. doi: 10.3389/fgene.2013.00049.

Polypay ewes

Methylomic and transcriptomic

Alfalfa haylage, corn, or dried corn distiller's grains

Namous H, Peñagaricano F, Del Corvo M, Capra E, Thomas DL, Stella A, Williams JL, Marsan PA, Khatib H. Integrative analysis of methylomic and transcriptomic data in fetal sheep muscle tissues in response to maternal diet during pregnancy. BMC Genomics. 2018 Feb 6;19(1):123. doi: 10.1186/s12864-018-4509-0. 

Polypay ewes

Gene expression changes in fetal muscle and adipose tissues

Alfalfa haylage, corn, or dried corn distiller's grains

Peñagaricano F, Wang X, Rosa GJ, Radunz AE, Khatib H. Maternal nutrition induces gene expression changes in fetal muscle and adipose tissues in sheep. BMC Genomics. 2014 Nov 28;15(1):1034. doi: 10.1186/1471-2164-15-1034.

Angus cross cows

Gene expression analysis

Low starch (LS; haylage) and high starch (HS; corn silage).

Wang X, Lan X, Radunz AE, Khatib H. Maternal nutrition during pregnancy is associated with differential expression of imprinted genes and DNA methyltranfereases in muscle of beef cattle offspring. J Anim Sci. 2015 Jan;93(1):35-40. doi: 10.2527/jas.2014-8148. 

Columbia/Rambouillet cross ewes

Expression of enzymes

Diet of 100% (Control) or 150% (Obese) of NRC

Long NM, Rule DC, Zhu MJ, Nathanielsz PW, Ford SP. Maternal obesity upregulates fatty acid and glucose transporters and increases expression of enzymes mediating fatty acid biosynthesis in fetal adipose tissue depots. J Anim Sci. 2012 Jul;90(7):2201-10. doi: 10.2527/jas.2011-4343. 

German Landrace sows

Gene expression and methylation

Dietary protein restriction

Altmann S, Murani E, Schwerin M, Metges CC, Wimmers K, Ponsuksili S. Maternal dietary protein restriction and excess affects offspring gene expression and methylation of non-SMC subunits of condensin I in liver and skeletal muscle. Epigenetics. 2012 Mar;7(3):239-52. doi: 10.4161/epi.7.3.19183.

Western White-faced ewes

Transcriptome (mRNA)

Control (100% NRC)-, restricted (60% NRC)-, or overfed (140% NRC)

Gauvin MC, Pillai SM, Reed SA, Stevens JR, Hoffman ML, Jones AK, Zinn SA, Govoni KE. Poor maternal nutrition during gestation in sheep alters prenatal muscle growth and development in offspring. J Anim Sci. 2020 Jan 1;98(1): skz388. doi: 10.1093/jas/skz388.

Comment 3:-Moreover, it this new knowledge applicable to goats for which species data are missing? Can the authors explain if there are benefits for that animal species as well?

Response 3: Although sheep were used as experimental animals in our experiment, the study question is valid for all mammals. The main question of the study is whether protein supplementation or restriction in pregnant women causes miRNA expression changes. Therefore, this question applies to all mammals, including goats.

Comment 4:-In describing the potential novelty in this study, please explain clearly the advancements achieved by this study over previously published papers.

Response 4:  As we mentioned above, no study similar to this study design has been found in any mammal species, including sheep. This study is the first in this design. The study we could find on maternal nutrition during pregnancy was also used in the discussion.

Comment 5:-The controls in the study are not clearly presented. Please add a new subsection to describe how you dealt with controls. Please note that explanations about positive and negative controls should be presented. Please describe accurately the controls for all stages of the study (animals, laboratory testing, material used, environmental conditions etc.).

Response 5:  All animals were exposed to same enviromental conditions in terms rearing. In the current study standard protein group were used as control group and high and low maternal protein intakes were compared accordingly. Neither positive nor negative control was used in the entire experiment. As you can see in the Table 1 in the manuscript, different protein groups were compared each other.

Comment 6: With regard to twin fetuses, please present the results in a separate group and make comparisons with the group of ewes with one fetus.

Response 6: In this study, it was not aimed to compare twins and singletons, but the effect of protein ratio in the maternal diet was investigated. Moreover, in nutrigenomic studies, pigs, mice, or rats are being used as multiparous model animals commonly which have large litter size. Although animals give different size of litters they are used for genetic analysis.  

Comment 7 -Comments on tables and figures. The tables are long and difficult to read. No figures have been included. Please move some tables to supplementary material. Please add 3 to 5 figures in order to make the results more easily understood. Please include photographs from the experimental work (e.g., gels or microarrays).

Response 7: Since graph visulizations seemed complicated, we prefered to give the results as table. Sample graph was given above. Tables were moved to supplementary data section. Tables 2-5 were given as Appendix and modified in the revised mansucript (headings were highligted with yellow color).

Comment 8 -The Discussion is a bit shallow and does not fully address all the results obtained. Please rewrite by extending it and by making reference to the specific findings for each stage of the study.

Response 8: Our findings contain:

  • Differentially expressed (DE) foetal miRNAs among different maternal protein intake groups. The main finding was miR-3957-5p which was DE in two groups. Biological evidence regarding this miRNA was given between lines 270-289. There we discussed about the previous studies both depending on our previos findings and literature. Moreover, miRNA cluster overlapping with this miRNA was given as well DLK1-DIO3 region which is crucial for muscle biology.
  • Genes targeted by miRNAs, detected in our study. Here we discussed in two ways a) single miRNAs, targeting group of genes b) multiple miRNAs, targeting specific genes. Especially multiple miRNAs targeting a gene is not commonly discussed in miRNAomics studies. This is discussed between the line 290-320.
  • Genes targeted by miRNAs, found in our study found to be related to immune functions according to pathway and GO analysis. These findings were discussed between lines 321-339.

Therefore, we believe that our findings discussed appropriately according to previous and current findings found in the literature.

Comment 9: Please include a paragraph about commercialisation of the findings. Is there a patent pending? You need to mention how this will be financed in clinical practice.

Response 9: Current study results do not have the potential to translate into a commercial product. The study was planned to reveal how excessive or inadequate protein nutrition during pregnancy affects the miRNA expression level in the offspring.

With the results of the study, unnecessary consumption of protein, the most valuable nutrient, can be prevented.

Comment 10: -Conclusions. Please avoid to introduce new ideas in Conclusions. Some of the Conclusions are a bit optimistic and not fully justified by findings. These need to be toned down.

Response 10: We modified conclusion section and rephared the last sentence.

Comment 11: What are the practical advantages of the findings for the average veterinarian in practice? How these findings can improve cash flow and larger clientele? Please add these in the revised version.

Response 11: With the results of the study, unnecessary consumption of protein, the most valuable nutrient, can be prevented.

Round 2

Reviewer 2 Report

Comments and Suggestions for Authors

I have no further comments.

Author Response

/First Academic Editor/

Comment: Thank you for submit your article and make the effort to address the concerns of all three reviewers. In my opinion, the articles contributes to the understanding of nutrigenomics in sheep.

Response: Thanks for your efforts to improve manuscript quality. Yes our aim was to investigate the effects of maternal protein intake on the fetal skeletal muscle miRNAome.

I have a few general comments:

Comment: Somewhere in the article, the authors need to clearly indicated that the reported effects occurred in the mid- to late- pregnancy, as the standard diet was fed for 30 days. What diets did ewes received from Day 105 to the end of gestation period? In addition, the authors need to clarify where the 15-day adaptation period fit into the pregnancy.

Response: Since fetuses were taken on the 105th day of gestation. Experiment was terminated and ewes were send back to research station for regular husbandry conditions. Since experiment was not continued, details were not mentioned in the manucript. Other revisions were done and highlighted with yellow color in the revised manuscript.

Comment: There are redundancies throughout the article and must be addressed. 

Simplify the sentences can at times make article easier to read and understand.

I have suggested some changes/rewrites in the attached file, hoping to make

the article easier to read and understand.

Response: Thanks for the file. We have also checked manuscript and all changings were highlighted as track changes in MS Word.

/Second Academic Editor/

Comment: The reviewer think you work lack a very important indicator, namely the

determination of the level of amino acids in the blood of mothers. Studying

gene expression is not enough to draw such profound conclusions. Could you

add these data?

Response: Thank you for good suggestion. When we designed the experiment, we have searched and read similar articles which we listed in previous revision letter. There, we have not seen any experiment that investigated maternal blood in terms of amino acids, glucose, fat etc.  

Reviewer 3 Report

Comments and Suggestions for Authors

The authors covered all the points. I have no further points.

Author Response

(The authors gave the same response as above.)
